**Subject Area:**
developmental biology/genetics/cellular biology/molecular biology/neuroscience

nucleotide excision repair, development, embryo, central nervous system, xeroderma pigmentosum, Cockayne syndrome

**Authors for correspondence:**
Sofia J. Araújo
e-mail: sofiajaraujo@ub.edu
Isao Kuraoka
e-mail: kuraoka@fukuoka-u.ac.jp

# Nucleotide excision repair genes shaping embryonic development

## Sofia J. Araújo[1,2] and Isao Kuraoka[3]

[1]Department of Genetics, Microbiology and Statistics, Faculty of Biology, University of Barcelona, 08028 Barcelona, Spain
[2]Institute of Biomedicine, University of Barcelona (IBUB), Barcelona, Spain
[3]Department of Chemistry, Faculty of Science, Fukuoka University, 8-19-1 Nanakuma, Jonan-ku, Fukuoka 814-0180, Japan

SJA, 0000-0002-4749-8913; IK, 0000-0001-6391-3411

Nucleotide excision repair (NER) is a highly conserved mechanism to remove helix-distorting DNA lesions. A major substrate for NER is DNA damage caused by environmental genotoxins, most notably ultraviolet radiation. Xeroderma pigmentosum, Cockayne syndrome and trichothiodystrophy are three human disorders caused by inherited defects in NER. The symptoms and severity of these diseases vary dramatically, ranging from profound developmental delay to cancer predisposition and accelerated ageing. All three syndromes include developmental abnormalities, indicating an important role for optimal transcription and for NER in protecting against spontaneous DNA damage during embryonic development. Here, we review the current knowledge on genes that function in NER that also affect embryonic development, in particular the development of a fully functional nervous system.

## 1. Human syndromes and NER deficiencies

The genome of all living beings exists in a dynamic equilibrium between ongoing DNA damage and reversal of the damage by DNA repair pathways. Multiple DNA repair mechanisms have evolved to shelter organisms from the continuous genotoxic stress induced by both intrinsic and extrinsic agents [1]. These agents can vary from cellular metabolites, such as reactive oxygen species (ROS), to environmental contaminants and ultraviolet (UV) radiation from the Sun [2]. DNA repair pathways can repair almost all possible DNA lesions created by these damaging agents. Consequently, a decrease in the cell's DNA repair capacity ultimately manifests itself in the form of mutagenesis, carcinogenesis, cellular senescence or cell death, and is implicated in a number of human diseases [3].

The disclosure of the intricacies of DNA repair has been made possible by the early description of human familial disease syndromes and by the more recent investigation of their genetic and molecular bases. The role of large protein complexes and the significance of their cellular localization are common features of many of the biochemical mechanisms involved. One of these DNA repair mechanisms is nucleotide excision repair (NER), which is responsible for removing a large variety of DNA lesions, including those helix-destabilizing DNA lesions induced by UV radiation [4]. There are two subclasses of NER. One is the global genome nucleotide excision repair (GG-NER), which removes lesions throughout the genome regardless of whether any specific sequence is transcribed or not. The other is the transcription-coupled nucleotide excision repair (TC-NER), which refers to the faster removal of damage from the transcribed strands of active genes.

Eukaryotic NER is a highly conserved multi-step process involving many different proteins whose molecular mechanism of action has been described in detail [5–9]. Alterations in NER genes are associated with autosomal

recessive human diseases, such as xeroderma pigmentosum (XP), Cockayne syndrome (CS) and trichothiodystrophy (TTD), whose symptoms involve skin cancer and developmental and neurological symptoms. Other human syndromes associated with mutations in proteins involved in NER are cerebro-oculo-facio-skeletal (COFS) syndrome, UV-sensitive syndrome (UVSS) and the rare combined XP/CS [10].

XP is a prototypical DNA repair disorder and is characterized by extreme sensitivity to UV light and a 2000-fold incidence in skin cancer. Patients who are severely affected by XP also experience late-onset neurological defects and some affected individuals have neurodevelopmental abnormalities [11]. In XP, the skin cancer-prone phenotype is readily explained by the inability of these patients to repair UV-induced DNA lesions in skin tissues exposed to sunlight. By contrast, patients with CS are not overly cancer prone, but they endure additional symptoms. CS is a multi-system disorder with pleiotropic effects and patients have severe neurological abnormalities (including myelination defects, calcification and microcephaly), mental retardation, growth and developmental abnormalities, lack of subcutaneous fat, hypogonadism, tooth decay, cataracts and shorter lifespans [12]. CS is also considered to be a premature ageing disorder with patients displaying progressive neurodegeneration [2]. TTD includes a spectrum of ectodermal abnormalities such as congenital ichthyosis, brittle hair and short stature. Some of the most affected patients have an increased incidence of skin cancers and a wide variety of central nervous system (CNS) abnormalities [13].

Seven complementation groups with defects in the NER pathway have been assigned genetically in XP (XP-A to XP-G). An eighth one, XP variant (XP-V), is proficient in NER, but carries mutations in the *POLH* gene, which encodes DNA polymerase $\eta$ (eta), a translesion synthesis (TLS) polymerase that specializes in error-free replication of DNA containing UV lesions [14,15].

The defining CS factors are Cockayne syndrome A (CSA) and B (CSB) proteins, although the CS phenotype can also result from specific mutations in some XP genes (*XPB*, *XPD* and *XPG*). In addition, another related factor, named XPA-binding protein 2 (XAB2), has been isolated as an XPA-interacting protein in a yeast two-hybrid screen. XAB2, a protein containing tetracopeptide repeats (TRP), also interacts with CSA, CSB and RNA polymerase II (RNAP2) [16]. Specifically, in cells treated with DNA-damaging agents, there was an enhanced interaction of XAB2 with RNAP2 or XPA [17]. Human cells depleted of XAB2 by RNAi show defects in transcription elongation and pre-mRNA splicing as well as hypersensitivity to killing by UV light and decreased recovery of RNA synthesis after UV irradiation, indicating that XAB2 is a multi-functional factor involved in splicing, transcription and TC-NER [17].

The transcription factor TFIIH is a central component of both NER processes (GG-NER and TC-NER). Mutations of its subunits are associated with both XP and CS. Like XAB2, TFIIH acts in distinct cellular processes. First, it is an essential component of the basic RNAP2 transcription machinery. Second, it is a basic DNA-repair factor, which is required for all repair by the NER pathway. And third, it can stimulate the ligand-dependent phosphorylation and activation of some nuclear receptors [18,19]. Genes for two subunits of TFIIH, *XPB* and *XPD*, are mutated in some

cases of XP and CS. XPG, another XP factor, is responsible for maintaining the integrity and function of TFIIH [18] and is involved in some forms of CS as well [20]. Hence, whereas XP is a disease more directly linked with the NER core reaction, CS is intrinsically connected with the transcriptional side of DNA repair and general transcription defects [21,22].

## 2. The NER reaction: global genome repair and transcription-coupled repair

Both GG-NER and TC-NER employ a common set of proteins but differ in their mode of DNA damage recognition. GG-NER requires detection of the damaged sites in DNA by the UV-damaged DNA-binding protein (UV-DDB) and a complex containing XP group C (XPC) protein, the human homologue of RAD23 (either of two paralogues RAD23A and RAD23B) and the centrosomal protein Centrin-2 (CETN2) [23–25]. As shown by cell-free systems and structural analysis, XPC interacts with damaged DNA and subsequently initiates the repair reaction [5,26,27]. Damage in the transcribed strand of active genes is repaired by TC-NER, which is initiated by a stalled RNAP2 during transcription and depends on recruitment of the ATP-dependent chromatin remodelling protein CS protein B (CSB) and the adaptor subunit for a CUL4A-based E3 ubiquitin ligase CS protein A (CSA) to the site of damage [19,28,29].

The NER reaction can be initiated by either of these two subpathways: GG-NER or TC-NER [30] (figure 1a). GG-NER can occur anywhere in the genome, whereas TC-NER is responsible for the accelerated repair of lesions in the transcribed strand of active genes. GG-NER is initiated by the GG-NER-specific factor XPC-RAD23B, in some cases with the help of UV-DDB [27]. TC-NER is initiated by RNAP2 stalled at a lesion with the help of TC-NER-specific factors CSA and CSB. Despite different beginnings, both pathways require the core NER factors to complete the excision process [10]. The core NER dual incision reaction has been reconstituted *in vitro* with purified factors using XPC-RAD23B, TFIIH, XPA, RPA, XPG and ERCC1-XPF [5]. Functional and structural studies revealed that XPC-RAD23B is the initial damage recognition factor in this system, as the presence of XPC-RAD23B is required for assembly of the other core NER factors and progression through the NER pathway both *in vitro* and *in vivo* [23,27,31,32].

The transcription and NER factor TFIIH is the next factor to join the NER complex and it is recruited by direct interaction with the XPC-RAD23B protein [2,33,34]. TFIIH consists of 10 subunits and can be divided up into the core (consisting of XPB, p52, p8, p62, p34, p44) and CAK (cyclin-activated kinase, consisting of CDK7, cyclin H and MAT1) complexes and the XPD protein that bridges the two [8]. The CAK complex dissociates from TFIIH and is not required for NER [5,35]. Of particular importance for the NER reaction are the two helicase subunits, XPB and XPD, which are known to open the DNA around the lesion [8,9,31]. The engagement of XPD with the lesion enables the full assembly of the pre-incision complex. XPA, RPA and XPG are next recruited to the site of the lesion independently of each other, and XPC-RAD23B departs from the complex at this point [36].

A central hub of the NER complex is XPA. It interacts with the TFIIH, RPA, XPC-RAD23B, DDB2, ERCC1-XPF and PCNA proteins, as well as with DNA. Through these

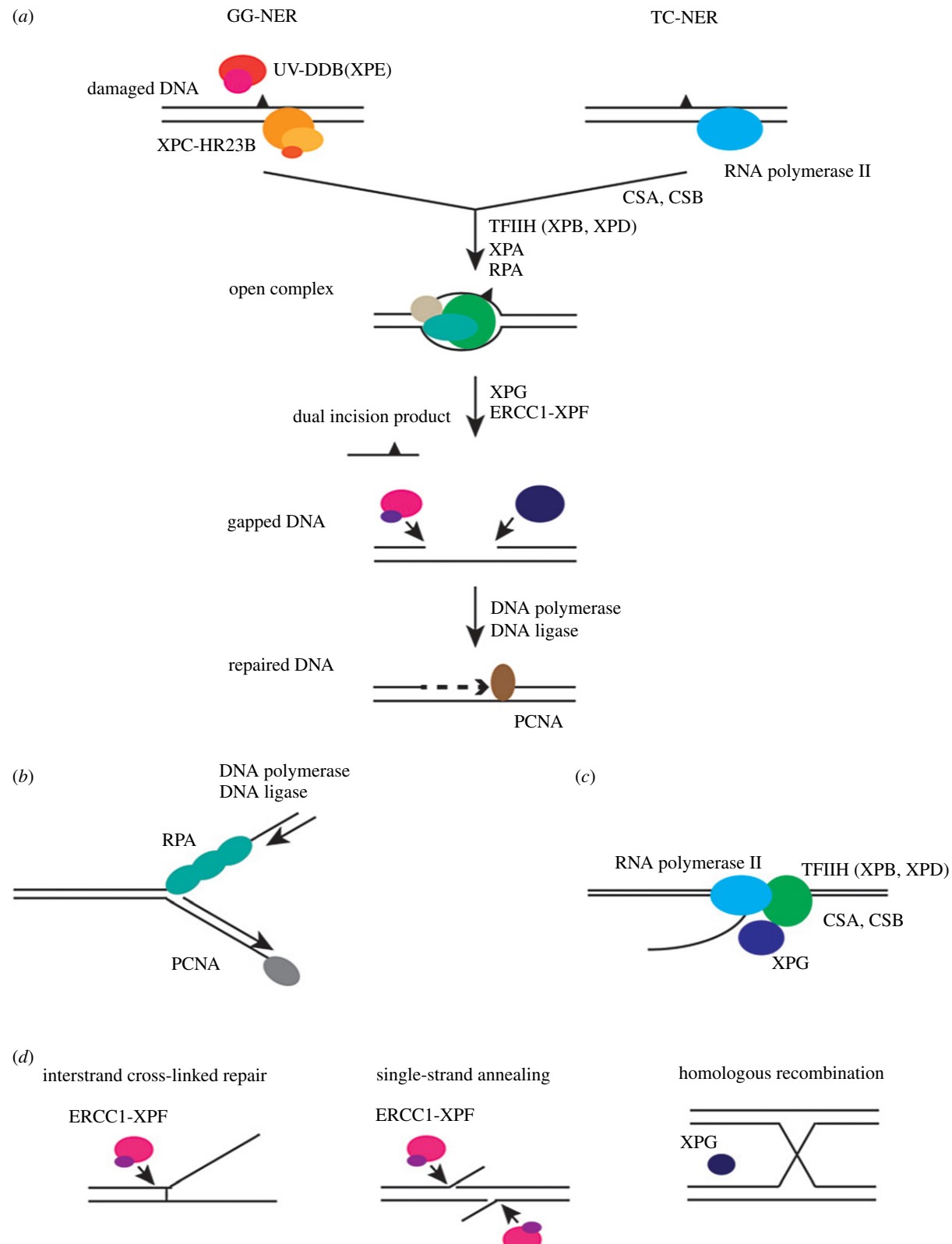

**Figure 1.** Schematic diagram of NER proteins involved in NER (TC-NER and GG-NER) and other pathways. (*a*) Schematic of the protein complexes involved in NER. Different recognition complexes operate during TC-NER and GG-NER. After the damage recognition step, the same protein complex is involved in damage excision and repair. NER factors also participate in replication (*b*), transcription (*c*) and other DNA repair pathways (*d*).

interactions, XPA occupies a central role as an NER factor and probably works to make sure that all the NER factors are in the right place for the incision to occur (reviewed in [30]).

XPA interacts tightly with the ssDNA-binding protein RPA in the NER complex and the two are believed to cooperate in their association with DNA. There is evidence that RPA binds the non-damaged DNA strand, helping position the two endonucleases ERCC1- XPF and XPG on

their substrate, the damaged DNA strand. The structure-specific endonuclease XPG is recruited through interaction with TFIIH. XPG, in fact, seems to be constitutively associated with TFIIH, at least for some of its roles in transcription [4,30,34]. Structural studies with recombinant human TFIIH show that XPB and XPD are stimulated by XPA and XPG and that these players change the mode of TFIIH from transcription to repair [9].

The complex consisting of TFIIH, XPA, RPA and XPG is relatively stable, and the dual excision reaction is only triggered once ERCC1-XPF joins the complex. ERCC1-XPF is recruited to NER complexes by interaction with the XPA protein. Once the two endonucleases are in place, dual incision at junctions between single-stranded and double-stranded DNA can be initiated [33,37]. Following the excision reaction, the lesion-containing oligonucleotide is released and the NER reaction finalizes with the resulting nucleotide single-stranded DNA gap being filled by DNA synthesis and ligation repair synthesis by DNA polymerases, associated factors and DNA ligase [5,30].

## 3. NER deficiencies and phenotype complexities

Many patients with mutations in NER or CS genes present developmental abnormalities at birth and may develop neurodegeneration later in life. Owing to the need for fast transcription during embryonic development [38,39] and in brain cells [40,41], many of these phenotypes may be due to the severely mutagenic and chromosome-destabilizing consequences of a stalled RNAP2. This could result in a transcriptional defect for critical genes, as well as a failure to accomplish TC-NER [42,43]. It has been hypothesized that TC-NER is more important for protecting non-dividing cells and neuronal function in the face of normal endogenous DNA damage [10,44]. This agrees with the general symptoms of XP-C patients, who have a defect in GG-NER but not in TC-NER and who present with neither developmental nor neurological abnormalities [45,46]. Interestingly, XP-A patients do not display obvious developmental phenotypes and do not seem to have widespread transcriptional impairment [47]. Affected individuals with mutations that completely ablate XPA function develop relatively normally, are born and may live for several decades. However, they often have various degrees of neurodegeneration [44]. Like other NER factors, XPA may have additional functions beyond NER. Recently, it was reported that XPA-deficient cells display mitochondrial dysfunction, with defects in mitophagy [46]. Mitochondrial dysfunction has been implicated in a number of pathophysiological processes such as ageing, neurodegenerative diseases, fertilization and embryonic development [48].

In fact, other NER factors are also involved not only in NER but also in replication, transcription and splicing (figure 1*b*,*c*). For instance, RPA was originally defined as a eukaryotic single-stranded DNA-binding protein essential for replication and an indispensable player in recombination (figure 1*b*). TFIIH is important for transcription initiation of RNAP2 during the expression of protein-coding genes and binds to a cyclin-activating kinase subcomplex for the cell cycle (figure 1*c*). Thus, the phenotypic complexity of patients with mutations in NER/CS genes might depend on a plethora of dysfunctional mechanisms (such as GG-NER, TC-NER, transcription, replication, recombination and splicing) fighting against DNA lesions in the context of the whole organism. In addition, we may speculate that some of the phenotype complexity could be due to neurodevelopment-specific DNA lesions recognized and repaired by NER. These still incompletely defined tissue-specific DNA lesions may have different effects on the organismal homeostasis.

In order to unravel the reason why NER-deficient patients develop neurodevelopmental abnormalities and neurodegeneration later in life, it is necessary to study possible embryonic-specific DNA lesions as well as which cellular mechanisms are impaired by them. A full understanding of the complex genotype/phenotype relationships of human DNA damage response disorders clearly requires further studies and suitable disease animal models [49,50].

## 4. NER and possible DNA lesions during embryonic development

As mentioned previously, human NER is the main pathway eliminating a wide variety of helix-destabilizing bulky DNA lesions that block DNA replication and transcription [1]. One important source of such DNA lesions is exposure to the UV component of sunlight, which generates photolesions (cyclobutane pyrimidine dimers (CPDs) and 6-4 pyrimidone photoproducts (6-4PPs)) in DNA. Cells from NER-deficient patients, that is, those with XP, CS or TTD, are extremely sensitive to UV light and patients with XP show an increased incidence of sunlight-induced skin cancers [2]. But what types of DNA damage may be responsible for the developmental abnormalities displayed by NER-deficient patients? UV radiation cannot generate photoinduced lesions in fetal or embryonic cells. So, sources of damage during development are most likely to be different, as NER eliminates not only UV-induced DNA lesions but also bulky DNA lesions such as the adducts induced by the anticancer drug cisplatin or mutagens like acetylaminofluorene [51]. Exposure to these carcinogenic substances may induce some of these NER-repairable lesions. However, these are neither very common nor a source of significant damage during human gestation.

Hence, DNA-damaging sources during embryonic development are most likely to be endogenous to cells, rather than exogenous. A spontaneous source of DNA damage inside patients' bodies is cellular generated ROS, such as superoxide and hydrogen peroxide, which produce hydroxyl radicals via the Fenton reaction that are highly reactive and cause various modified DNA bases [52]. Among them, 8-oxo-7,8-dihydroguanine (8-oxoG) is the most abundant and seems to play a major role in mutagenesis and in carcinogenesis. Interestingly, 8-oxoG is highly accumulated in the brain cells of patients with Alzheimer or Parkinson disease [53]. As a tissue, the brain is very sensitive to ROS, owing to its high oxygen consumption, about 20% of the whole body [54]. Thus, the brain is especially vulnerable to oxidative stress. In most cases, 8-oxoG is mainly removed from DNA by human base excision repair (BER) using 8-oxoguanine DNA glycosylase (OGG1), endonuclease III-like 1 (NTH1) and endonuclease VIII-like 1 (NEIL1) [55]. 8-oxoG is not a bulky, helix-destabilizing DNA lesion, but it has been reported that NER can also be involved in removing 8-oxoG from DNA [56].

Another important candidate for the endogenous generation of helix-distorting bulky DNA lesions by ROS is purine cyclodeoxynucleoside (cyPu) [52,57]. This type of lesion can block replication and it is unlikely to be removed by BER. Action of a glycosylase in BER would not be expected to release such cyPus, because the purine would remain attached by the 5′,8 carbon–carbon bond even after cleavage of the glycosyl bond. The cyPu lesions may be

royalsocietypublishing.org/journal/rsob    Open Biol. **9**: 190166

repaired by NER, which can remove oligonucleotides containing a DNA lesion by dual incision action. The lesions appear to be relatively abundant forms of DNA damage after exposure to ROS, introduced at 20–30% of the levels of the major lesions, although the relative rates of formation vary with experimental conditions. Thus, cyPu lesions in the brain might explain the progressive neurodegeneration seen in NER-deficient individuals [52,57]. Other candidate lesions for NER action are lipid peroxidation (LPO) product lesions and acetaldehyde-induced DNA lesions [58,59]. LPO products originate during normal cellular metabolism and generate protein and DNA adducts, which have detrimental effects in embryonic cells and can be repaired by NER [60,61]. Acetaldehyde is thought to cause a variety of DNA lesions and occurs naturally in various plants, ripe fruits and vegetables. In addition, drinking alcohol and smoking cigarettes can lead to high levels of acetaldehyde in the body that can be passed on to the developing fetus. Even without these environmental challenges, human cells are constantly exposed to acetaldehyde [58], and some acetaldehyde-induced DNA lesions might be repaired by NER. Interestingly, an acetaldehyde-GG cross-link resembles CPDs, 6-4PP and cisplatin-induced-GG adducts, and might be repaired as such. These lesions show an increase of GG-to-TT mutations in NER-deficient human XP cells [62]. Genome-wide analysis of sequence signatures indicates that GG-to-TT mutations are associated with cancer, suggesting that acetaldehyde in our body might induce DNA lesions [63]. During embryonic development, acetaldehyde can be detected in fetuses of alcoholic mothers and has been shown to have teratogenic effects [64,65].

By and large, it is unknown which kinds of DNA lesions cause developmental abnormalities in NER patients. Since NER, including both GG-NER and TC-NER, removes a wide variety of DNA lesions, it will be important to detect NER-repairable DNA lesions in cells during embryonic development.

## 5. NER and embryonic development

DNA repair is crucial both for dividing proliferating cells, in which lesions in DNA interfere with replication fork progression and may be converted into mutations upon replication, and for non-dividing differentiated cells, which sometimes have to maintain their genome integrity for the entire lifespan of the organism and have cell division-dependent checkpoints downregulated or switched off. In the first case, failure of DNA repair will induce mutations whereas in the second case it will give rise to an accumulation of DNA damage that can interfere with many cellular processes [45].

In actively proliferating cells, such as the cells of the early developing embryo, DNA repair is crucial for preventing the accumulation of mutations and synchronizing cell division [66,67]. Accordingly, it has been shown using the nematode *Caenorhabditis elegans* that early developmental stages are more sensitive to UV irradiation than later stages [68]. However, many developmental processes such as late organogenesis rely on fully differentiated cells, which are not actively dividing but frequently need to change their behaviours very rapidly, a process that relies on the fast transcription of many genes. Organ formation requires rapid cell proliferation, active gene transcription and a high rate of DNA metabolism, especially during the developmental

stages. Thus, embryonic cells are likely to be sensitive to both global-genome and transcribed-strand damage with slower rates of transcription leading to embryonic lethality [69]. In addition, an increase in NER capacity accompanies cell differentiation, as shown by the upregulated transcription of genes encoding XPA, XPC, XPG and ERCC1-XPF during neuron and muscle cell differentiation [70]. Hence, the proteins involved in the two NER pathways, GG-NER and TC-NER, are probably necessary for proper embryonic development, from the oocyte to fully developed organismal stages. Embryonic development can progress to term in the complete absence of NER, as shown by the apparently normal development and size of XPA knockout mice and humans [71]. Of course, there are severe developmental abnormalities displayed by many patients with XP or CS [72]. These are likely to be a combined effect of compromised transcription and DNA repair. In addition, if we consider that increased risk of developing cancer is based on intrinsic developmental defects at the molecular and cellular level, then most known DNA repair deficiencies are associated with significant developmental abnormalities.

As for many other biological studies, the ability to understand the interplay between NER and developmental processes requires appropriate model organisms. So far, much has been learned about human embryonic development and physiology through the study of model animals, which have particular advantages for laboratory research. There are many reasons for using them. Research on humans and other primates is expensive and limited by ethical considerations whereas the most commonly studied model animals are relatively inexpensive to maintain and are well suited for experimental manipulation [73]. In addition, recent research has shown that there is a remarkable degree of similarity in the developmental mechanisms of all animals. In developing model organism embryos, not only individual genes and proteins but also entire signalling pathways and cell behaviours appear highly conserved. This means that, although the embryology of simpler animals might appear superficially very different from that of humans, knowledge gained from those models can often be applied directly to understanding human developmental mechanisms. Furthermore, many of the known human disease-causing mutations are hypomorphic and animal models are the ideal way to study the effects of amorphic mutations during development, since many of these null mutants result in embryonic lethality. Therefore, research on the involvement of NER proteins in developmental biology has been largely done using model organisms.

## 6. Embryonic development without NER factors: survival and phenotypes

It has long been known that many NER proteins are actively expressed in many tissues during embryonic development even in the absence of external DNA-damaging agents [74–76]. NER genes, together with other DNA repair pathway genes, are expressed from early stages of embryonic development [77]. In widely studied model organisms such as *Mus musculus* or *Drosophila melanogaster*, expression of NER factors has been observed during development, ubiquitously in the whole organism or in specific tissues (table 1 and figure 2).

**Table 1.** Known embryonic transcript expression of NER transcripts in *Mus musculus* according to the Gene eXpression Database (GXD; http://www.informatics.jax.org/expression.shtml) and *Drosophila melanogaster* according to the Berkeley Drosophila Genome Project (BDGP; https://insitu.fruitfly.org). ND, not identified in this species; GUDMAP, GenitoUrinary Development Molecular Anatomy Project.

| mouse gene | embryonic expression | reference | *Drosophila* gene | embryonic expression | reference |
|---|---|---|---|---|---|
| *Xpa* | limb bud | [78] | *Xpac* | ventral nerve cord | [79] |
| *Xpb* | ubiquitous; nervous system and liver | [74] | *haywire* | ubiquitous | [80] and BDGP |
| *Xpc* | no expression data | — | *Xpc* | faint ubiquitous | [80] and BDGP |
| *Xpd* | nervous system, eye and liver | [81] | *Xpd* | ubiquitous (nuclear) | [82] |
| *Xpg* | nervous system | [83] | *mus201* | no available data | — |
| *Ercc1* | nervous system | [84] | *Ercc1* | no available data | — |
| *Ercc4* | nervous system | [83] | *Mei-9* | no available data | — |
| *mHR23B* | nervous system; genitourinary system | [81] and GUDMAP | *Rad23* | faint ubiquitous | [85] |
| *CSA (Ercc8)* | cranium | [86] | CSA (ND) | — | — |
| *CSB (Ercc6)* | genitourinary system | GUDMAP | CSB (ND) | — | — |
| *Xab2* | nervous system | [81] | *fandango* | ubiquitous | [38] |

One of the earliest observations of a strong influence of NER factors in embryonic development was a report showing that null mice lacking ERCC1 died before weaning [87]. Since then, other null mutations in NER genes have proven to be embryonic lethal in different species, suggesting a strong need for some of these factors during development [71] (table 2). When not lethal, many of these null mutations, such as *XPA* and *CSB*, induce growth retardation [75,105], another hint to their important functions during development (table 2). When some of these mutations are combined in the same animal, they give rise to stronger phenotypes, suggesting genetic interactions during developmental processes between many of these factors [71]. For instance, mice lacking both XPA and CSB displayed severe growth retardation, ataxia and motor dysfunction during early postnatal development, suggesting that these genes may have additive roles during nervous system development [106].

One of the crucial factors in NER is TFIIH, which is also one of the factors that bridges the two human syndromes XP and CS. Of the many TFIIH subunits, only XPB and XPD can be involved in both XP and CS. In *Drosophila*, loss of *haywire (hay)*, the gene homologous to XPB, leads to male sterility, CNS defects and UV sensitivity, not unlike human XPB/CS patients [89] (table 2). Hay is expressed in several stages of development and *hay* mutant embryos display phenotypes that range from completely disordered ventral nerve cords (VNCs) to VNCs with only a few broken commissures [89]. Transgenic flies carrying human-like alleles with mutations reported in human patients reproduce these defects, suggesting that *Drosophila* is a good model for these studies [107]. Another existing model for another TFIIH subunit, XPD, has been reported in *Drosophila*, allowing for different human mutations to be tested during development [108]. This *Drosophila* model revealed an Xpd function in cell cycle coordination which is affected by XP/CS and TTD mutations [108]. The two XP/CS alleles G47R and G675R, as well as the TTD allele R722 W, showed the highest frequency of asynchronous waves of all the *xpd* mutants in this *Drosophila* model. Human patients with these mutations display severe neurological abnormalities, reduced growth, and delayed and defective development, correlating the degree

of neurological abnormalities with asynchronous waves of cell division [108]. XPB and XPD mutants have also been analysed in other model organisms such as zebrafish or mouse (table 1). Overall, these two TFIIH subunits have been shown to be important for embryonic development across species [90,109]. XPB and XPD being subunits of TFIIH implies that their involvement in embryonic development is also due to their direct effects in transcription. Crippled transcription of key developmental genes might be responsible for the observed developmental phenotypes [110].

One more factor shown to be involved in both XP and CS is the endonuclease XPG [111]. Mice carrying truncated forms of XPG, generally associated with CS, exhibited postnatal growth failure and premature death, similar to the clinical hallmarks of CS despite apparent normal development [95]. In *Drosophila*, mutant flies are defective in NER and hypersensitive to UV radiation as the homozygous mutant mice. However, in contrast to these, the two *Drosophila* mutants are viable and fertile in the absence of exogenous DNA-damaging agents [112,113]. XPG has also been found to be a partner of BRCA1 and BRCA2 in maintaining genomic stability through homologous recombination (HRR) [114] (figure 1*d*). The role of this endonuclease in HRR suggests that this player has important roles in genome stability and may explain some of the phenotypes and clinical consequences associated with its loss of function.

The other NER endonuclease is ERCC1, which when mutated in mice leads to attenuated growth, resulting in cachectic dwarfism during the second week of life and premature death before postnatal day 35 [97]. This severe growth retardation was shown to originate from defective transcription initiation of developmental gene expression programmes [97]. In addition, ERCC1 has also been implicated in double-strand break, interstrand cross-link (ICL) and base excision repair [37] (figure 1*d*). This suggests that, as in the case of XPG, the developmental defects associated with mutations in ERCC1 may be due to transcriptional impairment as a consequence of faulty chromatin remodelling or other defective DNA damage responses, rather than to a direct effect of NER in the developmental programme. Interestingly, a metabolic connection was found between

royalsocietypublishing.org/journal/rsob Open Biol. 9: 190166

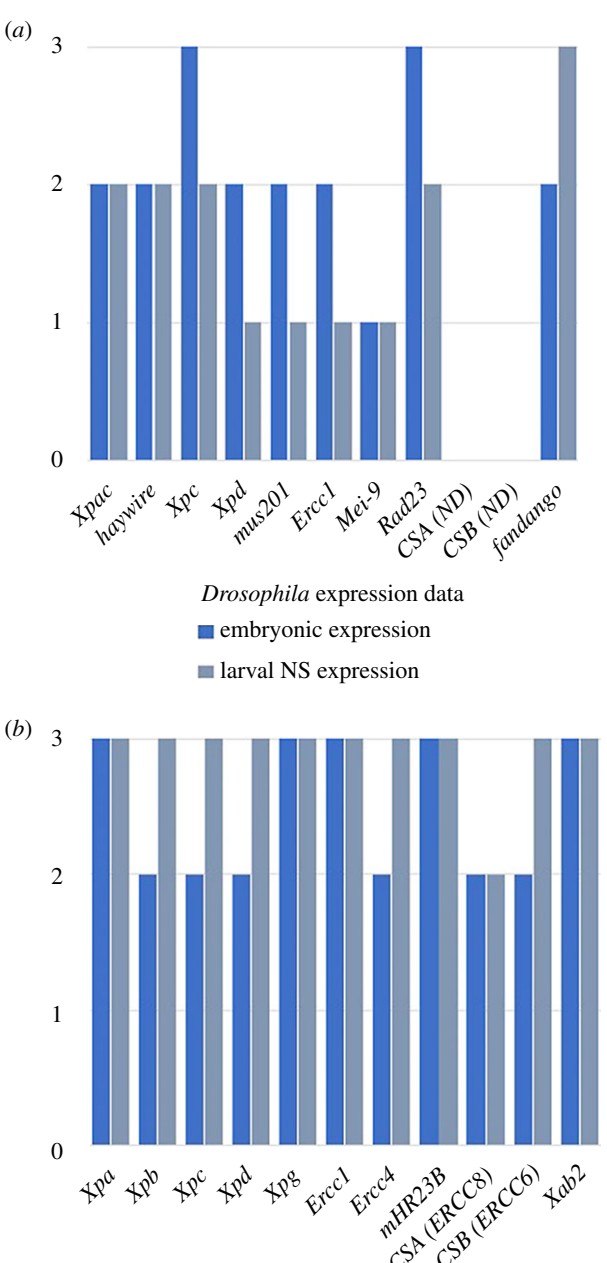

**Figure 2.** Embryonic and nervous system (NS) expression of NER genes in *Drosophila melanogaster* and *Mus musculus*. Graphical representation of transcript expression during embryonic development, according to high-throughput expression data. (*a*) *D. melanogaster* expression data from ModENCODE (www.modencode.org) tissue and temporal expression data; (*b*) *M. musculus* expression data from Expression Atlas (www.ebi.ac.uk/gxa/home) embryonic and tissue expression data. Data mining was performed according to all developmental stages (embryonic expression) and specific nervous system expression (larval expression for *Drosophila* and embryonic expression for mouse). Arbitrary values were attributed according to expression levels (1, low; 2, moderate; and 3, high) and plotted in parallel.

defects in ERCC1 and patients' phenotypes, suggesting an association between ERCC1 and organismal homeostasis and energy balance [115].

CSA (ERCC8) and CSB (ERCC6) are two factors directly associated with CS and mice deficient for either of these genetically mimic CS in humans [116]. However, when analysed at birth these mutants do not seem to show any developmental abnormalities, leading to the conclusion that CSA and CSB are not directly involved in any developmental process [116]. To gain further insight into these mutants and their effects in whole-organism homeostasis, various double mutant combinations were generated between CS and XP factors (reviewed in [116]). Of these, it is interesting to pinpoint the Csb/Xpa and Csb/Xpc double mutant mice, which had a very short lifespan and severe pathology in multiple tissues. In some litters, there was perinatal death and in others defects started very early in postnatal life. In addition, double mutant pups showed progressive development of ataxia and other motor dysfunctions, which correlated with smaller cerebella with a reduced number of granule cells [106]. In addition, Csb−/− embryonic and adult neural precursors exhibited defective self-renewal, and neurons differentiated *in vitro* from Csb−/− neural precursors, which had been irradiated with UV, exhibited defective neurite outgrowth [117]. Taken together, these data point at an active role of CSB during neurogenesis and the morphogenesis of the nervous system.

Irregularities in the regulation of transcription might account for many of the somatic features associated with CS, including neurological symptoms. CSB may have an important role in the transcriptional programmes that govern the plasticity and the maintenance of the CNS during early life [118]. Neurogenesis occurs both during embryonic development and later in life and failure to accomplish this process may lead to neurodevelopmental and neurodegeneration phenotypes. Accordingly, CSB deficiency has been shown to affect neuronal differentiation, suggesting that patients with CS are less able to support brain plasticity and repair events [119].

CS complementation genes *CSA* and *CSB* have also been studied in non-vertebrate models such as *C. elegans*. Mutations in the nematode *csa-1* and *csb-1* genes lead to developmental growth defects and UV sensitivity and both genes are expressed throughout embryonic development [120–122]. In *Drosophila*, neither *CSA* nor *CSB* homologues are present, despite their presence in many insect species [123]. It was reported in the past that repair of the transcribed strand occurs at the same speed as that of the non-transcribed strand both in embryonically derived cells and in brain tissue [124,125]. Lack of clear gene homology and biochemical data on GG versus TC-NER has led to the conclusion that *Drosophila* does not carry out TC-NER [126]. However, this is still under discussion, as flies would be the only model organism not to be able to actively repair highly transcribed genes. An alternative explanation is that there is CSB-independent TC-NER in *Drosophila* as has been shown in yeast [127,128]. Furthermore, the lack of differences between actively transcribed and non-transcribed genes in *Drosophila* was experimentally done using the *white* (*w*) gene as a control non-transcribed gene in both embryos and larval brains [124,125]. However, *w* expression could be detected in the same brain tissues where the comparison between repair of different strands was made [125]. And later reports have shown that, indeed, *w* is expressed both in embryos and in larvae and has pleiotropic effects in the whole organism [129,130]. Moreover, XAB2, a binding partner for CSA and CSB [17], has been recently identified in *Drosophila*, where it was named *fandango* (*fand*), and has been shown to be involved in embryonic pre-mRNA splicing and organogenesis [38,103]. As in *Drosophila*, null mutants for XAB2 in mice are embryonic lethal, pointing

royalsocietypublishing.org/journal/rsob  Open Biol. 9: 190166

**Table 2.** Requirements of NER genes during development inferred by the analysis of null mutations in *Mus musculus* and *Drosophila melanogaster*.

| mouse gene | phenotype | *Drosophila* gene | phenotype | reference |
|---|---|---|---|---|
| *Xpa* | viable; develop normally | *Xpac* | no mutant developmental data | [88] |
| *Xpb* | embryonic lethal | *haywire* | embryonic lethal; CNS defects | [89,90] |
| *Xpc* | viable; develop normally | *Xpc* | no mutant developmental data | [91,92] |
| *Xpd* | pre-implantation lethality | *Xpd* | embryonic lethal; early mitotic division defects | [93,94] |
| *Xpg* | mice are viable but die before weaning | *mus201* | no mutant developmental data | [95,96] |
| *Ercc1* | viable but growth failure and death before weaning | *Ercc1* | no mutant developmental data | [87,97] |
| *Ercc4* | severe postnatal growth defect with death approximately three weeks after birth | *Mei-9* | no mutant developmental data | [98,99] |
| *mHR23B* | impaired embryonic development; prenatal and early postnatal death (90%) | *Rad23* | no mutant developmental data | [100] |
| *CSA (Ercc8)* | viable; minor postnatal growth retardation and neurological defects | not identified | — | [101] |
| *CSB (Ercc6)* | viable; minor postnatal growth retardation and neurological defects | not identified | — | [102] |
| *Xab2* | embryonic lethal | *fandango* | embryonic lethal; organogenesis defects | [103,104] |

at the important function of this gene during development [104]. So, the quest for factors controlling possible TC-NER in *Drosophila* is still on.

Overall, all current data seem to point out that many factors involved in NER are also important during embryonic development. However, during these studies, analysis of developmental defects was done without the challenge of exogenous DNA repair, during normal development, taking into account only endogenous levels of DNA damage. Hence, the effects of NER/CS mutations in development are mostly analysed under conditions that mimic low levels of DNA damage. Stronger phenotypes are attained if embryos are subjected to exogenous DNA damage. A study using *C. elegans* has revealed that DNA ICLs lead to developmental arrest and tissue defects in mutants for NER proteins [131], revealing the importance of NER in embryos subjected to extra sources of DNA damage.

Taken all together, the role of NER in embryonic development is not yet well understood; however, a number of clues have surfaced indicating that efficient repair of endogenous damage may be crucial to normal development. It seems that factors involved in GG-NER as well as TC-NER are required for proper development. The close association between RNA transcription and the repair of bulky lesions on the transcribed strand of the DNA suggests that efficient repair of lesions that block transcription is crucial for sustaining the complex cellular balance required for proper development. Hence, it is likely that this aspect of NER is essential for proper development.

We have come a long way, but much more information is needed to determine to what extent NER of endogenous or environmentally induced DNA damage is influential during the correct formation of an organism and what are the cross-talks between the NER machinery and the developmental programmes.

Data accessibility. This article has no additional data.
Authors' contributions. S.J.A. and I.K. conceived and wrote the paper.
Competing interests. We declare we have no competing interests.
Funding. This collaborative work was supported by the Ministry of Education, Culture, Sports, Science and Technology (MEXT) of Japan (Grant-in-Aid for Scientific Research (B) 26650006) and by the Central Research Institute of Fukuoka University (no. 1810310).

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
