## [Reviewer comments · Open Biology]

Review History

RSOB-19-0166.R0 (Original submission)

Review form: Reviewer 1

Recommendation

Major revision is needed (please make suggestions in comments)

Do you have any ethical concerns with this paper?

No

Comments to the Author

Araújo and Kuraoka

This is a reasonable start on a review about a very interesting area, authored by two experts. I recommend that it is thoroughly revised and updated with focus on the following three points.

1. A major theme is not coming across clearly: many mammalian NER proteins have dual

functions, and many of them are essential. I know that the authors understand this point well, but it is not coming across in the article. Only mild defects in some of the NER proteins can be tolerated. The developmental defects in XPB, XPD for example are most likely related to sub-optimal transcription of some genes rather than NER defects. Similarly, ERCC1 and XPF are essential genes that function in SSA, the FA pathway, etc.

There is one figure (Fig 1) on NER, but it does not seem directly relevant to the theme of the review, "NER in development". At least one additional figure could be included to make the point of multifunctional proteins that affect overall transcription, or other repair pathways, as well as NER.

2. The citation list seems focused on the 1990s, early 2000's, and there has been quite a lot learned that could be useful here. For example, read through these and see how to incorporate into the thinking:

Recent cryoEM structures of TFIIF in transcriptional mode and NER mode show how the proteins are disposed in these processes, where XPA binds, where nucleases cut, and suggest transition to the next steps in transcription & repair.

Kokic G, Chernev A, Tegunov D, Dienemann C, Urlaub H, Cramer P (2019) Structural basis of TFIIF activation for nucleotide excision repair. *Nat Commun* 10: 2885. PMID:31253769
<https://www.ncbi.nlm.nih.gov/pubmed/31253769>

Greber BJ, Toso DB, Fang J, Nogales E (2019) The complete structure of the human TFIIF core complex. *Elife* 8: PMID:30860024
<https://www.ncbi.nlm.nih.gov/pubmed/30860024>

XPG is an essential gene, a paper exploring the phenotypes is here. There may be homologous recombination defects as suggested here.

Trego KS, Groesser T, Davalos AR, Parpys AC, Zhao W, Nelson MR, Hlaing A, Shih B, Rydberg B, Pluth JM et al. (2016) Non-catalytic Roles for XPG with BRCA1 and BRCA2 in Homologous Recombination and Genome Stability. *Mol Cell* 61: 535-546. PMID:26833090
<https://www.ncbi.nlm.nih.gov/pubmed/26833090>

Although it has been proposed that all the NER proteins might affect transcription, this has been looked at with XPA. A defect in XPA only affects a limited number of transcripts in human cells. This first paper is worth a look, as well as the *C. elegans* and mouse studies cited therein. Note in the second it has been proposed that XPA defects cause mitochondrial defects as well. How would this affect development?

Manandhar M, Lowery M, Boulware K, Lin K, Lu Y, Wood R (2017) Transcriptional Consequences of XPA Disruption in Human Cell Lines. *DNA Repair* 57: 76-90. PMID:28704716
<http://www.sciencedirect.com/science/article/pii/S1568786417301258>

Fang EF, Scheibye-Knudsen M, Brace LE, Kassahun H, SenGupta T, Nilsen H, Mitchell JR, Croteau DL, Bohr VA (2014) Defective mitophagy in XPA via PARP-1 hyperactivation and NAD(+)/SIRT1 reduction. *Cell* 157: 882-96. PMID:24813611
<http://www.ncbi.nlm.nih.gov/pubmed/24813611>

Recognition by XPC and DDB is better understood:
<https://academic.oup.com/nar/article/47/12/6015/5490814>

Quite a lot is known about TCR recognition steps CSA, CSB and UVSSA. For example:
<https://www.biorxiv.org/content/10.1101/707216v1>

CSB defective cells also have general transcription defects, not just TC-NER, see Newman, J.C., Bailey, A.D. and Weiner, A.M. (2006) Cockayne syndrome group B protein (CSB)

plays a general role in chromatin maintenance and remodeling. *Proc Natl Acad Sci U S A*, 103, 9613-9618; Lake, R.J. and Fan, H.Y. (2013) Structure, function and regulation of CSB: a multi-talented gymnast. *Mech Ageing Dev*, 134, 202-211.

I think that XAB2 is probably irrelevant, I think it's never been shown to be an NER factor or IP directly with XPA? It's an RNA splicing factor which is why it is essential.

Defects in ERCC1-XPF have major effects on energy balance. There are neurological defects in the hippocampus of these mice (and XPG deficient). Dietary restriction helps rescue the mice! This is an amazing and wonderful paper:

Vermeij WP, Dolle ME, Reiling E, Jaarsma D, Payan-Gomez C, Bombardieri CR, Wu H, Roks AJ, Botter SM, van der Eerden BC et al. (2016) Restricted diet delays accelerated ageing and genomic stress in DNA-repair-deficient mice. *Nature* 537: 427-431. PMID:27556946

<https://www.ncbi.nlm.nih.gov/pubmed/27556946>

Other ERCC1-XPF info

Gregg SQ, Robinson AR, Niedernhofer LJ (2011) Physiological consequences of defects in ERCC1-XPF DNA repair endonuclease. *DNA Repair (Amst)* 10: 781-91. PMID:21612988

<http://www.ncbi.nlm.nih.gov/pubmed/21612988>

Manandhar M, Boulware KS, Wood RD (2015) The ERCC1 and ERCC4 (XPF) genes and gene products. *Gene* 569: 153-161. PMID:26074087 <http://www.ncbi.nlm.nih.gov/pubmed/26074087>

More is known about endogenous sources of damage that could be NER-relevant.

Lipid peroxidation is one that should be considered, causing adducts:

Czerwinska J, Nowak M, Wojtczak P, Dziuban-Lech D, Ciesla JM, Kolata D, Gajewska B, Baranczyk-Kuzma A, Robinson AR, Shane HL et al. (2018) ERCC1-deficient cells and mice are hypersensitive to lipid peroxidation. *Free Radic Biol Med* 124: 79-96. PMID:29860127

<https://www.ncbi.nlm.nih.gov/pubmed/29860127>

Yousefzadeh MJ, Zhu Y, McGowan SJ, Angelini L, Fuhrmann-Stroissnigg H, Xu M, Ling YY, Melos KI, Pirtskhalava T, Inman CL et al. (2018) Fisetin is a senotherapeutic that extends health and lifespan. *EBioMedicine* 36: 18-28. PMID:30279143

<https://www.ncbi.nlm.nih.gov/pubmed/30279143>

3. For a review of this type it would be best to include some original insight about NER during development (that's the title ...). For example, I recommend that the authors do some data mining on specific questions, and plot a few interesting graphs or tables.

Are the NER genes expressed differentially during development? In different tissues of an embryo? There is a lot of data out there. Can it help explain specific neurological defects?

Taking XPA as an example

e.g. Developmental expression of genes,

Allen Brain Atlas

http://search.brain-map.org/search/index.html?query=%40entrez_id%2022590

<http://mouse.brain-map.org/experiment/show?id=70474743>

Expression Atlas:

https://www.ebi.ac.uk/gxa/genes/ensmusg00000028329?bs=%7B%22mus%20musculus%22%3A%5B%22ORGANISM_PART%22%5D%7D&ds=%7B%22kingdom%22%3A%5B%22animals%22%5D%7D#baseline

Using GXD expression Atlas:
<https://www.ncbi.nlm.nih.gov/pubmed/24958384>

Decision letter (RSOB-19-0166.R0)

17-Sep-2019

Dear Dr Araújo:

We are writing to inform you that the Editor has reached a decision on your manuscript RSOB-19-0166 entitled "Nucleotide Excision Repair during embryonic development", submitted to Open Biology.

As you will see from the reviewers' comments below, there are a number of criticisms that prevent us from accepting your manuscript at this stage. The reviewers suggest, however, that a revised version could be acceptable, if you are able to address their concerns. If you think that you can deal satisfactorily with the reviewer's suggestions, we would be pleased to consider a revised manuscript.

The revision will be re-reviewed, where possible, by the original referees. As such, please submit the revised version of your manuscript within six weeks. If you do not think you will be able to meet this date please let us know immediately.

When submitting your revised manuscript, please respond to the comments made by the referee(s) and upload a file "Response to Referees" in "Section 6 - File Upload". You can use this to document any changes you make to the original manuscript. In order to expedite the processing of the revised manuscript, please be as specific as possible in your response to the referee(s).

Please see our detailed instructions for revision requirements
<https://royalsociety.org/journals/authors/author-guidelines/>

Sincerely,

Open Biology
mailto: openbiology@royalsociety.org

Reviewer(s)' Comments to Author(s):

Referee: 1

Comments to the Author(s)

Araújo and Kuraoka

This is a reasonable start on a review about a very interesting area, authored by two experts. I recommend that it is thoroughly revised and updated with focus on the following three points.

1. A major theme is not coming across clearly: many mammalian NER proteins have dual functions, and many of them are essential. I know that the authors understand this point well, but it is not coming across in the article. Only mild defects in some of the NER proteins can be tolerated. The developmental defects in XPB, XPD for example are most likely related to sub-optimal transcription of some genes rather than NER defects. Similarly, ERCC1 and XPF are essential genes that function in SSA, the FA pathway, etc.

There is one figure (Fig 1) on NER, but it does not seem directly relevant to the theme of the review, "NER in development". At least one additional figure could be included to make the point of multifunctional proteins that affect overall transcription, or other repair pathways, as well as NER.

2. The citation list seems focused on the 1990s, early 2000's, and there has been quite a lot learned that could be useful here. For example, read through these and see how to incorporate into the thinking:

Recent cryoEM structures of TFIIH in transcriptional mode and NER mode show how the proteins are disposed in these processes, where XPA binds, where nucleases cut, and suggest transition to the next steps in transcription & repair.

Kokic G, Chernev A, Tegunov D, Dienemann C, Urlaub H, Cramer P (2019) Structural basis of TFIIH activation for nucleotide excision repair. *Nat Commun* 10: 2885. PMID:31253769
<https://www.ncbi.nlm.nih.gov/pubmed/31253769>

Greber BJ, Toso DB, Fang J, Nogales E (2019) The complete structure of the human TFIIH core complex. *Elife* 8: PMID:30860024 <https://www.ncbi.nlm.nih.gov/pubmed/30860024>

XPG is an essential gene, a paper exploring the phenotypes is here. There may be homologous recombination defects as suggested here.

Trego KS, Groesser T, Davalos AR, Parplys AC, Zhao W, Nelson MR, Hlaing A, Shih B, Rydberg B, Pluth JM et al. (2016) Non-catalytic Roles for XPG with BRCA1 and BRCA2 in Homologous Recombination and Genome Stability. *Mol Cell* 61: 535-546. PMID:26833090
<https://www.ncbi.nlm.nih.gov/pubmed/26833090>

Although it has been proposed that all the NER proteins might affect transcription, this has been looked at with XPA. A defect in XPA only affects a limited number of transcripts in human cells. This first paper is worth a look, as well as the *C. elegans* and mouse studies cited therein. Note in the second it has been proposed that XPA defects cause mitochondrial defects as well. How would this affect development?

Manandhar M, Lowery M, Boulware K, Lin K, Lu Y, Wood R (2017) Transcriptional Consequences of XPA Disruption in Human Cell Lines. *DNA Repair* 57: 76-90. PMID:28704716
<http://www.sciencedirect.com/science/article/pii/S1568786417301258>

Fang EF, Scheibye-Knudsen M, Brace LE, Kassahun H, SenGupta T, Nilsen H, Mitchell JR, Croteau DL, Bohr VA (2014) Defective mitophagy in XPA via PARP-1 hyperactivation and NAD(+)/SIRT1 reduction. *Cell* 157: 882-96. PMID:24813611
<http://www.ncbi.nlm.nih.gov/pubmed/24813611>

Recognition by XPC and DDB is better understood:
<https://academic.oup.com/nar/article/47/12/6015/5490814>

Quite a lot is known about TCR recognition steps CSA, CSB and UVSSA. For example:
<https://www.biorxiv.org/content/10.1101/707216v1>

CSB defective cells also have general transcription defects, not just TC-NER, see Newman, J.C., Bailey, A.D. and Weiner, A.M. (2006) Cockayne syndrome group B protein (CSB) plays a general role in chromatin maintenance and remodeling. *Proc Natl Acad Sci U S A*, 103, 9613-9618; Lake, R.J. and Fan, H.Y. (2013) Structure, function and regulation of CSB: a multi-talented gymnast. *Mech Ageing Dev*, 134, 202-211.

I think that XAB2 is probably irrelevant, I think it's never been shown to be an NER factor or IP directly with XPA? It's an RNA splicing factor which is why it is essential.

Defects in ERCC1-XPF have major effects on energy balance. There are neurological defects in the hippocampus of these mice (and XPG deficient). Dietary restriction helps rescue the mice! This is an amazing and wonderful paper:

Vermeij WP, Dolle ME, Reiling E, Jaarsma D, Payan-Gomez C, Bombardieri CR, Wu H, Roks AJ, Botter SM, van der Eerden BC et al. (2016) Restricted diet delays accelerated ageing and genomic stress in DNA-repair-deficient mice. *Nature* 537: 427-431. PMID:27556946
<https://www.ncbi.nlm.nih.gov/pubmed/27556946>

Other ERCC1-XPF info

Gregg SQ, Robinson AR, Niedernhofer LJ (2011) Physiological consequences of defects in ERCC1-XPF DNA repair endonuclease. *DNA Repair (Amst)* 10: 781-91. PMID:21612988
<http://www.ncbi.nlm.nih.gov/pubmed/21612988>

Manandhar M, Boulware KS, Wood RD (2015) The ERCC1 and ERCC4 (XPF) genes and gene products. *Gene* 569: 153-161. PMID:26074087 <http://www.ncbi.nlm.nih.gov/pubmed/26074087>

More is known about endogenous sources of damage that could be NER-relevant.

Lipid peroxidation is one that should be considered, causing adducts:

Czerwinska J, Nowak M, Wojtczak P, Dziuban-Lech D, Ciesla JM, Kolata D, Gajewska B, Baranczyk-Kuzma A, Robinson AR, Shane HL et al. (2018) ERCC1-deficient cells and mice are hypersensitive to lipid peroxidation. *Free Radic Biol Med* 124: 79-96. PMID:29860127
<https://www.ncbi.nlm.nih.gov/pubmed/29860127>

Yousefzadeh MJ, Zhu Y, McGowan SJ, Angelini L, Fuhrmann-Stroissnigg H, Xu M, Ling YY, Melos KI, Pirtskhalava T, Inman CL et al. (2018) Fisetin is a senotherapeutic that extends health and lifespan. *EBioMedicine* 36: 18-28. PMID:30279143
<https://www.ncbi.nlm.nih.gov/pubmed/30279143>

3. For a review of this type it would be best to include some original insight about NER during development (that's the title ...). For example, I recommend that the authors do some data mining on specific questions, and plot a few interesting graphs or tables.

Are the NER genes expressed differentially during development? In different tissues of an embryo? There is a lot of data out there. Can it help explain specific neurological defects?

Taking XPA as an example

e.g. Developmental expression of genes,

Allen Brain Atlas

http://search.brain-map.org/search/index.html?query=%40entrez_id%2022590

<http://mouse.brain-map.org/experiment/show?id=70474743>

Expression Atlas:

https://www.ebi.ac.uk/gxa/genes/ensmusg00000028329?bs=%7B%22mus%20musculus%22%3A%5B%22ORGANISM_PART%22%5D%7D&ds=%7B%22kingdom%22%3A%5B%22animals%22%5D%7D#baseline

Using GXD expression Atlas:

<https://www.ncbi.nlm.nih.gov/pubmed/24958384>

Author's Response to Decision Letter for (RSOB-19-0166.R0)

See Appendix A.

RSOB-19-0166.R1 (Revision)

Review form: Reviewer 1

Recommendation

Accept with minor revision (please list in comments)

Do you have any ethical concerns with this paper?

No

Comments to the Author

This is an up-dated and thoughtful revision from two experts. It is certainly a major improvement on the original. I like the addition of Figure 2 and the Tables, this adds considerable interest. Some more comments for attention in improving the presentation are below:

1. Most important, statement on page 12 cannot be correct: "Hence, the two NER pathways, GG-NER and TC-NER, are probably necessary for proper embryonic development, from the oocyte to fully developed organismal stages."

We know this is not right because XPA patients have zero NER, and their embryonic development is normal. Same for XPA mice, humans, flies, C. elegans etc. No fitness decrease or decrease in litter size. (see also author's Table 2) The XPA-associated deficits come in the form of accelerated neurological deterioration later in life.

Later in the article, the discussion is good and more nuanced, complete and scholarly with respect to other genes with dual functions in NER and other processes.

2. ERCC6 and ERCC8 genes, are definitely both missing from *Drosophila* (Sekelsky et al. 2000a). ERCC6 is missing from the whole Dipteran order of insects (and some nonarthropod phyla), whereas ERCC8 cannot be found in any Holometabola.
<https://www.genetics.org/content/205/2/471>

Perhaps there is CSB-independent TC-NER in *Drosophila*.

In yeast there is Rad26-independent TCR
<https://www.ncbi.nlm.nih.gov/pubmed/10666451?dopt=Citation>

It is dependent on Rpb9
<https://www.ncbi.nlm.nih.gov/pubmed/12411509>

See also discussion in "DNA Repair and Mutagenesis" 2nd edition book.

3. XPV, I recommend that instead of citing only the Hanaoka 1999 lab reference, a review for XPV – pol eta mutations to avoid omitting the Prakash papers and other related info, see ref 13 Lehmann et al that I used in News and Views, or something similar
<https://www.nature.com/articles/d41586-018-05255-1>

4. Page 7,
 "many of these phenotypes may be due to the severely mutagenic and chromosome-destabilizing consequences of a stalled RNAP2 and a failure to accomplish TC-NER (40, 41)."

I recommend that you also include the possibility that some of the neurological phenotypes might arise from a transcriptional defect.

5. Page 9 and elsewhere: note that CS and TTD patients do not show an increased incidence of cancers. Only XP.

6. Page 9, hydrogen peroxide and superoxide do not react much if at all with DNA. They generate hydroxyl radicals (.OH) via an iron-catalyzed Fenton reaction which are highly reactive and then damage DNA.

Decision letter (RSOB-19-0166.R1)

27-Sep-2019

Dear Dr Araújo

We are pleased to inform you that your manuscript RSOB-19-0166.R1 entitled "Nucleotide Excision Repair and its players during embryonic development" has been accepted by the Editor for publication in *Open Biology*. The reviewer(s) have recommended publication, but also suggest some minor revisions to your manuscript. Therefore, we invite you to respond to the reviewer(s)' comments and revise your manuscript.

Please submit the revised version of your manuscript within 14 days. If you do not think you will be able to meet this date please let us know immediately and we can extend this deadline for you.

- 1) A text file of the manuscript (doc, txt, rtf or tex), including the references, tables (including captions) and figure captions. Please remove any tracked changes from the text before submission. PDF files are not an accepted format for the "Main Document".
- 2) A separate electronic file of each figure (tiff, EPS or print-quality PDF preferred). The format should be produced directly from original creation package, or original software format. Please note that PowerPoint files are not accepted.
- 3) Electronic supplementary material: this should be contained in a separate file from the main text and meet our ESM criteria (see <http://royalsocietypublishing.org/instructions-authors#question5>). All supplementary materials accompanying an accepted article will be treated as in their final form. They will be published alongside the paper on the journal website and posted on the online figshare repository. Files on figshare will be made available approximately one week before the accompanying article so that the supplementary material can be attributed a unique DOI.

Online supplementary material will also carry the title and description provided during submission, so please ensure these are accurate and informative. Note that the Royal Society will not edit or typeset supplementary material and it will be hosted as provided. Please ensure that the supplementary material includes the paper details (authors, title, journal name, article DOI). Your article DOI will be 10.1098/rsob.2016[last 4 digits of e.g. 10.1098/rsob.20160049].

- 4) A media summary: a short non-technical summary (up to 100 words) of the key findings/importance of your manuscript. Please try to write in simple English, avoid jargon, explain the importance of the topic, outline the main implications and describe why this topic is newsworthy.

Images

Data-Sharing

It is a condition of publication that data supporting your paper are made available. Data should be made available either in the electronic supplementary material or through an appropriate

repository. Details of how to access data should be included in your paper. Please see <http://royalsocietypublishing.org/site/authors/policy.xhtml#question6> for more details.

Data accessibility section

Sincerely,
The Open Biology Team
<mailto:openbiology@royalsociety.org>

Reviewer(s)' Comments to Author:

Referee: 1

Comments to the Author(s)

This is an up-dated and thoughtful revision from two experts. It is certainly a major improvement on the original. I like the addition of Figure 2 and the Tables, this adds considerable interest. Some more comments for attention in improving the presentation are below:

1. Most important, statement on page 12 cannot be correct: "Hence, the two NER pathways, GG-NER and TC-NER, are probably necessary for proper embryonic development, from the oocyte to fully developed organismal stages."

We know this is not right because XPA patients have zero NER, and their embryonic development is normal. Same for XPA mice, humans, flies, *C. elegans* etc. No fitness decrease or decrease in litter size. (see also author's Table 2) The XPA-associated deficits come in the form of accelerated neurological deterioration later in life.

Later in the article, the discussion is good and more nuanced, complete and scholarly with respect to other genes with dual functions in NER and other processes.

2. ERCC6 and ERCC8 genes, are definitely both missing from *Drosophila* (Sekelsky et al. 2000a). ERCC6 is missing from the whole Dipteran order of insects (and some nonarthropod phyla), whereas ERCC8 cannot be found in any Holometabola.

<https://www.genetics.org/content/205/2/471>

Perhaps there is CSB-independent TC-NER in *Drosophila*.

In yeast there is Rad26-independent TCR
<https://www.ncbi.nlm.nih.gov/pubmed/10666451?dopt=Citation>

It is dependent on Rpb9
<https://www.ncbi.nlm.nih.gov/pubmed/12411509>

See also discussion in "DNA Repair and Mutagenesis" 2nd edition book.

3. XPV, I recommend that instead of citing only the Hanaoka 1999 lab reference, a review for XPV – pol eta mutations to avoid omitting the Prakash papers and other related info, see ref 13 Lehmann et al that I used in News and Views, or something similar <https://www.nature.com/articles/d41586-018-05255-1>

4. Page 7,

“many of these phenotypes may be due to the severely mutagenic and chromosome-destabilizing consequences of a stalled RNAP2 and a failure to accomplish TC-NER (40, 41).”

I recommend that you also include the possibility that some of the neurological phenotypes might arise from a transcriptional defect.

5. Page 9 and elsewhere: note that CS and TTD patients do not show an increased incidence of cancers. Only XP.

6. Page 9, hydrogen peroxide and superoxide do not react much if at all with DNA. They generate hydroxyl radicals (.OH) via an iron-catalyzed Fenton reaction which are highly reactive and then damage DNA.

Author's Response to Decision Letter for (RSOB-19-0166.R1)

See Appendix B.

Decision letter (RSOB-19-0166.R2)

30-Sep-2019

Dear Dr Araújo

We are pleased to inform you that your manuscript entitled "Nucleotide Excision Repair genes shaping embryonic development" has been accepted by the Editor for publication in Open Biology.

Sincerely,

The Open Biology Team
mailto: openbiology@royalsociety.org

Appendix A

20th of September, 2019

Open Biology RSOB-19-0166

Nucleotide Excision Repair and its players during embryonic development

Dear Prof. David Glover,

Thank you for your letter and for the opportunity of submitting a new revised version of our manuscript.

We are now resubmitting a substantially revised and improved manuscript. We have updated our bibliography and provide new data-mining analysis, new figure panels and tables. In addition, we have changed many parts of the text and also our title in order to be more clear about the information revised.

Below, we include a response to the reviewers' comments in bold dark blue.

We hope you will find that our answers have satisfactorily addressed your concerns and that you will consider that the new manuscript is appropriate for publication in Open Biology.

Sincerely,

Sofia J. Araújo and Isao Kuraoka

Referee #1

Araújo and Kuraoka

This is a reasonable start on a review about a very interesting area, authored by two experts. I recommend that it is thoroughly revised and updated with focus on the following three points.

1. A major theme is not coming across clearly: many mammalian NER proteins have dual functions, and many of them are essential. I know that the authors understand this point well, but it is not coming across in the article. Only mild defects in some of the NER proteins can be tolerated. The developmental defects in XPB, XPD for example are most likely related to sub-optimal transcription of some genes rather than NER defects. Similarly, ERCC1 and XPF are essential genes that function in SSA, the FA pathway, etc.

There is one figure (Fig 1) on NER, but it does not seem directly relevant to the theme of the review, "NER in development". At least one additional figure could be included to make the point of multifunctional proteins that affect overall transcription, or other repair pathways, as well as NER.

We have completed figure 1 with more details of the involvement of NER factors in replication, transcription and other repair pathways, which might also influence development.

2. The citation list seems focused on the 1990s, early 2000's, and there has been quite a lot learned that could be useful here. For example, read through these and see how to incorporate into the thinking:

Recent cryoEM structures of TFIIH in transcriptional mode and NER mode show how the proteins are disposed in these processes, where XPA binds, where nucleases cut, and suggest transition to the next steps in transcription & repair.

Kokic G, Chernev A, Tegunov D, Dienemann C, Urlaub H, Cramer P (2019) Structural basis of TFIIH activation for nucleotide excision repair. *Nat Commun* 10: 2885. PMID:31253769

<https://www.ncbi.nlm.nih.gov/pubmed/31253769>

Greber BJ, Toso DB, Fang J, Nogales E (2019) The complete structure of the human TFIIH core complex. *Elife* 8: PMID:30860024

<https://www.ncbi.nlm.nih.gov/pubmed/30860024>

XPG is an essential gene, a paper exploring the phenotypes is here. There may be homologous recombination defects as suggested here.

Trego KS, Groesser T, Davalos AR, Parplys AC, Zhao W, Nelson MR, Hlaing A, Shih B, Rydberg B, Pluth JM et al. (2016) Non-catalytic Roles for XPG with BRCA1 and BRCA2 in Homologous Recombination and Genome Stability. *Mol Cell* 61: 535-546.

PMID:26833090 <https://www.ncbi.nlm.nih.gov/pubmed/26833090>

Although it has been proposed that all the NER proteins might affect transcription, this has been looked at with XPA. A defect in XPA only affects a limited number of transcripts in human cells. This first paper is worth a look, as well as the *C. elegans* and mouse studies cited therein. Note in the second it has been proposed that XPA defects cause mitochondrial defects as well. How would this affect development?

Manandhar M, Lowery M, Boulware K, Lin K, Lu Y, Wood R (2017) Transcriptional Consequences of XPA Disruption in Human Cell Lines. *DNA Repair* 57: 76-90.

PMID:28704716 <http://www.sciencedirect.com/science/article/pii/S1568786417301258>

Fang EF, Scheibye-Knudsen M, Brace LE, Kassahun H, SenGupta T, Nilsen H, Mitchell JR, Croteau DL, Bohr VA (2014) Defective mitophagy in XPA via PARP-1 hyperactivation and NAD(+)/SIRT1 reduction. *Cell* 157: 882-96. PMID:24813611

<https://www.ncbi.nlm.nih.gov/pubmed/24813611>

Recognition by XPC and DDB is better understood:

<https://academic.oup.com/nar/article/47/12/6015/5490814>

Quite a lot is known about TCR recognition steps CSA, CSB and UVSSA. For example:

<https://www.biorxiv.org/content/10.1101/707216v1>

CSB defective cells also have general transcription defects, not just TC-NER, see Newman, J.C., Bailey, A.D. and Weiner, A.M. (2006) Cockayne syndrome group B protein (CSB) plays a general role in chromatin maintenance and remodeling. *Proc Natl Acad Sci U S A*, **103**, 9613-9618; Lake, R.J. and Fan, H.Y. (2013) Structure, function and regulation of CSB: a multi-talented gymnast. *Mech Ageing Dev*, **134**, 202-211.

I think that XAB2 is probably irrelevant, I think it's never been shown to be an NER factor or IP directly with XPA? It's an RNA splicing factor which is why it is essential.

We have changed all text references of XAB2 being a TC-NER factor. However, we still include it due to its probable importance during development.

Defects in ERCC1-XPF have major effects on energy balance. There are neurological defects in the hippocampus of these mice (and XPG deficient). Dietary restriction helps rescue the mice! This is an amazing and wonderful paper:

Vermeij WP, Dolle ME, Reiling E, Jaarsma D, Payan-Gomez C, Bombardieri CR, Wu H, Roks AJ, Botter SM, van der Eerden BC et al. (2016) Restricted diet delays accelerated ageing and genomic stress in DNA-repair-deficient mice. *Nature* 537: 427-431. PMID:27556946 <https://www.ncbi.nlm.nih.gov/pubmed/27556946>

Other ERCC1-XPF info

Gregg SQ, Robinson AR, Niedernhofer LJ (2011) Physiological consequences of defects in ERCC1-XPF DNA repair endonuclease. *DNA Repair (Amst)* 10: 781-91. PMID:21612988 <http://www.ncbi.nlm.nih.gov/pubmed/21612988>

Manandhar M, Boulware KS, Wood RD (2015) The ERCC1 and ERCC4 (XPF) genes and gene products. *Gene* 569: 153-161. PMID:26074087 <http://www.ncbi.nlm.nih.gov/pubmed/26074087>

More is known about endogenous sources of damage that could be NER-relevant.

Lipid peroxidation is one that should be considered, causing adducts:

Czerwinska J, Nowak M, Wojtczak P, Dziuban-Lech D, Ciesla JM, Kolata D, Gajewska B, Baranczyk-Kuzma A, Robinson AR, Shane HL et al. (2018) ERCC1-deficient cells and mice are hypersensitive to lipid peroxidation. *Free Radic Biol Med* 124: 79-96. PMID:29860127 <https://www.ncbi.nlm.nih.gov/pubmed/29860127>

Yousefzadeh MJ, Zhu Y, McGowan SJ, Angelini L, Fuhrmann-Stroissnigg H, Xu M, Ling YY, Melos KI, Pirtskhalava T, Inman CL et al. (2018) Fisetin is a senotherapeutic that extends health and lifespan. *EBioMedicine* 36: 18-28. PMID:30279143 <https://www.ncbi.nlm.nih.gov/pubmed/30279143>

We appreciate this wonderful and thorough update on our bibliography. We have updated the manuscript to include data from these publications in the main text and added these references to our list.

3. For a review of this type it would be best to include some original insight about NER *during* development (that's the title ...). For example, I recommend that the authors do some data mining on specific questions, and plot a few interesting graphs or tables.

Are the NER genes expressed differentially during development? In different tissues of an embryo? There is a lot of data out there. Can it help explain specific neurological defects?

Taking XPA as an example

e.g. Developmental expression of genes,

Allen Brain Atlas

http://search.brain-map.org/search/index.html?query=%40entrez_id%2022590

<http://mouse.brain-map.org/experiment/show?id=70474743>

Expression Atlas:

<https://www.ebi.ac.uk/gxa/genes/ensmusg00000028329?bs=%7B%22mus%20musculus%22%3A%5B%22ORGANISM%20PART%22%5D%7D&ds=%7B%22kingdom%22%3A%5B%22animals%22%5D%7D#baseline>

Using GXD expression Atlas:

<https://www.ncbi.nlm.nih.gov/pubmed/24958384>

We analysed the general and NS embryonic NER transcript expression in both Drosophila and mouse according to the curated data in Flybase/Flyexpress and GXD, respectively. We have also analysed high-throughput expression data in the same model organisms. This data-mining analysis was plotted in a new figure (Fig.2).

Appendix B

Answer to reviewer's comments

Review

This is an up-dated and thoughtful revision from two experts. It is certainly a major improvement on the original. I like the addition of Figure 2 and the Tables, this adds considerable interest. Some more comments for attention in improving the presentation are below:

1. Most important, statement on page 12 cannot be correct: "Hence, the two NER pathways, GG-NER and TC-NER, are probably necessary for proper embryonic development, from the oocyte to fully developed organismal stages."

We know this is not right because XPA patients have zero NER, and their embryonic development is normal. Same for XPA mice, humans, flies, *C. elegans* etc. No fitness decrease or decrease in litter size. (see also author's Table 2) The XPA-associated deficits come in the form of accelerated neurological deterioration later in life.

Later in the article, the discussion is good and more nuanced, complete and scholarly with respect to other genes with dual functions in NER and other processes.

We have changed the above-mentioned sentence to include NER proteins/genes/factors instead of the NER process, as such.

2. ERCC6 and ERCC8 genes, are definitely both missing from *Drosophila* (Sekelsky et al. 2000a). ERCC6 is missing from the whole Dipteran order of insects (and some nonarthropod phyla), whereas ERCC8 cannot be found in any Holometabola. <https://www.genetics.org/content/205/2/471>

Perhaps there is CSB-independent TC-NER in *Drosophila*.

In yeast there is Rad26-independent TCR

<https://www.ncbi.nlm.nih.gov/pubmed/10666451?dopt=Citation>

It is dependent on Rpb9

<https://www.ncbi.nlm.nih.gov/pubmed/12411509>

See also discussion in "DNA Repair and Mutagenesis" 2nd edition book.

ERCC6 and ERCC8 homologues cannot be found in *Drosophila*, but there might be a functional homologue responsible for TC-NER in this organism. We have included the reference to Rpb9 dependent Rad26 independent TC-NER as an alternative explanation for the lack of clear homologues in *Drosophila*.

3. XPV, I recommend that instead of citing only the Hanaoka 1999 lab reference, a review for XPV – pol eta mutations to avoid omitting the Prakash papers and other related info, see ref 13 Lehmann et al that I used in News and Views, or something similar <https://www.nature.com/articles/d41586-018-05255-1>

We have included this reference.

4. Page 7,

“many of these phenotypes may be due to the severely mutagenic and chromosome-destabilizing consequences of a stalled RNAP2 and a failure to accomplish TC-NER (40, 41).”

I recommend that you also include the possibility that some of the neurological phenotypes might arise from a transcriptional defect.

We have changed this sentence to “This could result in a transcriptional defect for critical genes, as well as a failure to accomplish TC-NER (42, 43).”

5. Page 9 and elsewhere: note that CS and TTD patients do not show an increased incidence of cancers. Only XP.

We have revised the manuscript to be more clear about this.

6. Page 9, hydrogen peroxide and superoxide do not react much if at all with DNA. They generate hydroxyl radicals ($\cdot\text{OH}$) via an iron-catalyzed Fenton reaction which are highly reactive and then damage DNA.

We have clarified this part.